# Cu-Doped Porous Carbon Derived from Heavy Metal-Contaminated Sewage Sludge for High-Performance Supercapacitor Electrode Materials

**DOI:** 10.3390/nano9060892

**Published:** 2019-06-17

**Authors:** Zhouliang Tan, Feng Yu, Liu Liu, Xin Jia, Yin Lv, Long Chen, Yisheng Xu, Yulin Shi, Xuhong Guo

**Affiliations:** 1Key Laboratory for Green Processing of Chemical Engineering of Xinjiang Bingtuan, School of Chemistry and Chemical Engineering, Shihezi University, Shihezi 832003, China; tzl6880@foxmail.com (Z.T.); yufeng05@mail.ipc.ac.cn (F.Y.); liuliu66shzu@163.com (L.L.); jiaxin_shzu@foxmail.com (X.J.); ag_125@163.com (Y.L.); chenlong2012@sinano.ac.cn (L.C.); ecustn@gmail.com (Y.X.); guoxuhong@ecust.edu.cn (X.G.); 2State Key Laboratory of Chemical Engineering, East China University of Science and Technology, Shanghai 200237, China

**Keywords:** heavy metal, flocculation, Cu-doped carbon, supercapacitor

## Abstract

In this paper, we report a complete solution for enhanced sludge treatment involving the removal of toxic metal (Cu(II)) from waste waters, subsequent pyrolytic conversion of these sludge to Cu-doped porous carbon, and their application in energy storage systems. The morphology, composition, and pore structure of the resultant Cu-doped porous carbon could be readily modulated by varying the flocculation capacity of Cu(II). The results demonstrated that it exhibited outstanding performance for supercapacitor electrode applications. The Cu(II) removal efficiency has been evaluated and compared to the possible energy benefits. The flocculant dosage up to 200 mg·L^−1^ was an equilibrium point existing between environmental impact and energy, at which more than 99% Cu(II) removal efficiency was achieved, while the resulting annealed product showed a high specific capacity (389.9·F·g^−1^ at 1·A·g^−1^) and good cycling stability (4% loss after 2500 cycles) as an electrode material for supercapacitors.

## 1. Introduction

The heavy metal contamination resulting from mining, smelting activities, exhaust gas discharge, and sewage irrigation has detrimental effects on human health and environmental sustainability. Recent studies have consistently shown an association between heavy metal pollution and physical discomfort, and even death, in humans [1]. However, because of their poor degradability, dissolved heavy metal ions normally need to be physically removed from wastewater by immobilization techniques such as ion exchange, flocculation, membrane filtration, and adsorption [2]. Even though these approaches aid the removal of heavy metals from wastewaters, they generate large quantities of sewage sludge. It is estimated that over 8 × 10^6^ tons of sewage sludge are generated each day from wastewater treatment plants in China [3],which takes further issue in sludge handling and final disposal.

As an important residue produced from wastewater treatment, sewage sludge recovery and recycling are always considered as an adequate challenge in wastewater management [1]. Sewage sludge landfill, incineration processes, soil application, and dumping at sea are considered as the most effective methods for the disposal of waste sludge in many countries due to high availability and relatively low costs [4]. As for China, more than 80% of sludge was dumped improperly, followed by sanitary landfill (13.4%), building materials (2.4%), incineration (0.36%), and land application (0.24%) [5]. However, these techniques are highly controversial due to serious secondary pollution from leachate and air emissions, which is reflected in the significantly high heavy-metal levels found in the vicinity. Therefore, some authors have stressed that traditional wastewater treatment will only shift the problem from water contamination to soil and air pollution, and the potential risks of heavy metals could be greatly reduced, but cannot be entirely eliminated [6]. It is a fact that sewage treatment plants and relevant enterprises are still struggling with rising costs from sludge transport and disposal, and the seeking of the integration planning between environmental protection and sustainable development has become increasingly pronounced.

More recently, attempts have been made to transform environmental waste products into advanced energy conversion and storage materials through direct thermal disposal of sewage sludge [7]. This is particularly crucial for economic and social development since it offers both environmental and energy benefits. Supercapacitors, as new energy storage devices, have garnered great attention in recent years [8]. Carbon-based materials are one of the most widely used electrodes for supercapacitors with desired physical and chemical characterizations [9]. However, they constantly demonstrate lower capacitance and inferior energy density. The use of electrode materials possessing pseudo-capacitance is vital to ensure high capacity as well as high energy and power density [10]. Hence, one of the most significant approaches in the fabrication of high-capacitance materials is the incorporation of transition metals into carbon frameworks. Previous research has demonstrated an obvious enhancement in the electrochemical performance of carbon materials through metal-doping (Mn [11], Ni [12], Co [13], Fe [14]), which can store and release energy reversibly through surface redox reactions and display considerably higher specific capacitance through an additional pseudocapacitive storage contribution between the electro-active materials and the electrolyte [15]. However, expensive metal-doped precursors were used, which makes the synthetic processes costly and difficult to scale-up [16]. Thus, the generation of a low-cost and abundant metal-containing precursor to fabricate high-quality electrode materials is extremely desirable.

Most sewage sludge are naturally rich in carbon (derived from organic polymer flocculants) and heavy metals (e.g., Mn, Cu, Zn, Pb, Cr, and Cd) [17]. Utilizing this heavy metal-contaminated sewage sludge as precursors, metal-doped carbon materials can be prepared on a large scale and at a relatively low cost without any additional metal salts. In addition, some heavy metal oxides can store and release energy reversibly through surface redox reactions between the electro-active materials and the electrolyte, which usually display considerably higher specific capacitance through an additional pseudocapacitive storage contribution [15]. More importantly, the heavy metals would be confined within the carbon frameworks by the complete pyrolysis of sludge, which effectively eliminates the transference of toxic heavy metal. To the best of our knowledge, very few research efforts have been made so far to synthesize metal-doped carbon materials from heavy metal-contaminated sewage sludge. In addition, a number of studies have reported that the preparation and electrochemical performances of electrode materials are sensitive to the composition and structure of the precursor [18]. Generally, the composition of sewage sludge (flocculants and metal) could be easily controlled by the flocculation coefficient during the wastewater treatment process [19]. Therefore, the recycling of waste sludge in an environmentally and economically acceptable way would involve wastewater treatment processes in the early stage.

The main aim of this paper was to study the effects of flocculation conditions on physical properties and pyrolysis products of sludge floc. While dissolved metal pollutants can be removed efficiently from wastewaters by means of the coagulation–flocculation process, they also generate large quantities of toxic sewage sludge. Therefore, we developed a facile, cost-effective, and green approach to synthesize a series of Cu-doped carbon materials through direct carbonization of Cu(II)-containing sludge and their application in energy storage systems (Scheme 1). A starch-based flocculant containing ionizable carboxyl group was synthesized and used as a promising flocculant for heavy metal ion (Cu(II)) flocculation from wastewaters. The effects of flocculant dose and pH were studied for the flocculation of Cu(II) from wastewater using batch studies mode. Sewage sludge with different Cu(II) flocculation capacity were employed as direct precursors for the synthesis of Cu-doped carbon materials. The influence of Cu(II) flocculation capacity on the pyrolysis of sewage sludge and electrochemical performances of electrode material was investigated. The effect of metal content, morphology, and pore structure of the resultant electrode materials on the electrochemical energy storage properties were discussed. Finally, the Cu(II) removal efficiency has also been evaluated relative to possible energy storage benefits, as well as to cost and purification efficiency from wastewater treatment operations.

## 2. Materials and Methods

### 2.1. Materials

Raw maize starch (food-grade, purity >98%, *w*/*w*) was purchased from Heng-hui Food Co., Ltd. (Xinjiang, China). Glycine (99% pure), polyvinylidene fluoride (PVDF, 10 wt %), and 1-methyl-2-pyrrolidone were provided by Shanghai Maclean Biochemical Technology Co., Ltd. (Shanghai, China). Ammonium hydroxide (purity >25–28%, *w*/*w*) and CuCl_2_•2H_2_O (99% pure) were obtained from Fuyu Fine Chemical Co., Ltd. (Tianjin, China). Cyanuric chloride (99% pure) was obtained from the Three Character fine chemical Co., Ltd. (Liaoning, China). Starch-based flocculants (SF) were synthesized using 2-chloro-4,6-diglycino-[1,3,5]-triazine as the etherification agent (Details in Appendix A). Distilled H_2_O was used for all experiments.

### 2.2. Flocculation Experiment

Before each assessment, a flocculant solution of 4 g·L^−1^ was obtained by dissolving a certain amount of SF in distilled water. CuCl_2_•2H_2_O was dissolved in distilled water to afford the Cu(II) stock solution of 4 g·L^−1^, the pH of which was adjusted to 8.1 using NH_3_•H_2_O.

Cu(II) flocculation experiments were performed using batch assessments at room temperature. Specifically, 0.6, 1.4, 1.6, 2, 2.4, and 2.6 mL of flocculant solution were added to a glass beaker (100 mL) containing 1.2 mL Cu(II) stock solution at pH 8.1. After that, water was added to obtain a total volume of 40 mL. Finally, Cu (II) concentration was 120 mg L^−1^. The suspension was initiated by rapidly stirring at 300 rpm for 5 min, and then slowly mixed for 10 min at 80 rpm. The precipitate was allowed to stand for 30 min and the solution was filtered (2 μm filter paper) to obtain the filtrate for residual concentration (RC) assessments. RC of Cu(II) was estimated from the derived calibration curve (Appendix A). Jar tests were repeated at least three times, the results of which were analyzed with mean values and standard deviation. Furthermore, 2.4 mL flocculant solution was added to a glass beaker (100 mL) containing 1.2 mL Cu(II) stock solution at pH 3, 3.5, 4, 4.5, 5, 5.5, 7, and 8.5, respectively. Water was added to obtain a total volume of 40 mL. After that, the same steps above was used to conduct the remaining flocculation experiments and detection of Cu (II).

Cu(II) removal (R%; Equation (1)) and flocculation capacity (Q; Equation (2)) could be calculated as follows:(1)$$R=\frac{C_{0}-C_{f}}{C_{0}}\times 100\%$$
(2)$$Q=V\frac{C_{0}-C_{f}}{m}$$
in which *C*_0_ and *C_f_* (mg·L^−1^) are the initial and final concentrations of Cu(II) in the filtrate, respectively. *V* (mL) represents the volume of the solution, and *m* (mg) represents the dried mass of SF. Flocs with different Cu(II) flocculation capacities (*Q* = 0.21–0.90 mg·mg^−1^, *R* = 45.1–99.5%) were collected and labeled as SF-x (x = *Q* = 0.21–0.90 mg·mg^−1^).

### 2.3. Fabrication of Cu-Doped Carbon Materials Using Cu(II)-Containing Sludge

Cu-doped carbon materials (SFC) were prepared from Cu(II)-containing sludge with various flocculation capacities, i.e., 0.9 (*R* = 45.0%), 0.6 (*R* = 99.5%), and 0.25 (*R* = 45.7%) mg·mg^−1^, respectively. Typically, Cu(II)-containing sludge as a precursor was pyrolyzed (1 h at 300 °C and 2 h at 800 °C) under an Ar atmosphere at a rate of 5 °C min^−1^. After prolonged cooling down to room temperature, the dark powder produced was dispersed in H_2_O while stirring to eliminate ash and other inorganic compounds, and then denoted as SFC-x (x = 0.9, 0.6, and 0.25).

### 2.4. Characterization Methods

The Cu(II) concentration was determined using Inductively Coupled Plasma-Atomic Emission Spectrometry (Cu: 324.754 nm, ICP-AES, Varian 710E S). The zeta potential (ZP) was assessed using a nanoplus zeta/nano particle analyzer (Otsuka Electronics) and the optical transmittance was determined by means of a UV-visible spectrophotometer (UV-6100S, METASH, Shanghai, China). X-ray diffraction (XRD, Rigaku TTR III) was used for analyzing the sample structure using Cu Kα radiation. Sample morphology was imaged using a scanning electron microscope (SEM, JEOL JSM-6490LV). Transmission electron microscopy (TEM) and high-resolution transmission electron microscopy (HRTEM) were performed using a FEI Tecnai G2 (FEI Company, Hillsboro, OR, USA). X-ray photoelectron spectroscopic (XPS) measurements were conducted on a PHI-5000C (Physical Electronics, Inc. (PHI), Chanhassen, MN, USA). A quantachrome Autosorb surface analyzer (Quantachrome Instruments, Boynton Beach, FL, USA) was used to perform BET surface area measurements at 77.3 K.

### 2.5. Electrochemical Measurements

Working electrodes were prepared via mixing active materials (80 wt %), carbon (10 wt %), and polyvinylidene fluoride (PVDF, 10 wt %) in 1-methyl-2-pyrrolidone to form a slurry, which was then brushed onto nickel foam (active area: 1 × 1 cm^2^) and dried at 60 °C for 24 h. The active material (weight: 5 mg), a Pt foil (1 × 1 cm^2^), and saturated calomel electrodes (SCE) were used as working, counter, and reference electrodes, and the electrolyte was 6 M potassium hydroxide (KOH). The electrochemical performance was estimated by cyclic voltammetry (CV), galvanostatic charge/discharge (GCD) measurement, and electrochemical impedance spectroscopy (EIS) on a CHI 760E analyzer (CH Instruments Inc., Shanghai, China). Gravimetric specific capacitance (Csp, F·g^−1^) for single electrodes was calculated from each galvanostatic charge/discharge curve as follows (Equation (3)):(3)$$Cs=\frac{I\times ∆t}{m\times ∆V}$$
where *m*: the mass of the active material, *I*: discharge current, ∆*V*: potential change after complete discharge, and ∆*t*: time for complete discharge.

## 3. Results

### 3.1. Removal of Cu(II) from Aqueous Solution

Visible transmittance change of the flocculant dispersions was recorded for pH values from 3 to 9 at 25 °C (Figure 1a). Clearly, such a starch-based flocculant shows a unique pH-responsive phase transition behavior. As a carboxyl group-functionalized starch derivative, the carboxyl groups get progressively deprotonated at high pH, which enhances the overall hydrophilic property of the flocculant, resulting in a significant increase in solution transmittance [20]. As can be seen from Figure 1a, the flocculant could easily dissolve in water to yield a clear solution with pH ≥8, indicating that a reversible equilibrium was reached with respect to the deprotonation/protonation of the carboxyl group. Zeta potential results also confirmed that the flocculant exhibited a typical anionic property in the investigated pH range of 2.5–9.0 (Figure 1b). The surface charge of the colloidal particles changed from a point of zero charge around pH 2.4 to a highly negative value of −38 mV at pH 8.1. In order to determine the optimum conditions for heavy metal ion removal, the flocculation of Cu(II) was investigated by varying the parameters of flocculation pH and flocculant dosage in simulated wastewater.

The influence of pH on the copper removal is depicted in Figure 1c. With an increase in the pH of the flocculation solution, flocculation performance was considerably improved and the Cu(II) removal efficiency reached 99.1% at pH 8.5. As mentioned above, the solubility and zeta potential of the flocculant is greatly affected by the deprotonation of the carboxyl groups and the degree of deprotonation of the carboxyl groups increases with increasing pH [21]. Therefore, heavy metal cations can be eliminated from aqueous media via a combination of charge neutralization and polymer bridging [22].

To investigate the effect of the flocculant dosage on Cu(II) removal, various dosages (60–260 mg·L^−1^) were exposed to a fixed Cu(II) concentration (120 mg·L^−1^) at pH 8.1. In general, a low flocculant dosage with simultaneously high heavy metal removal efficiency is greatly desirable for industrial wastewater treatment. An optimal flocculant dosage not only reduces flocculation cost but also decreases the total quantity of sewage sludge, since the flocculant utilization can be reduced to a minimum during wastewater treatment. The effects of flocculant dose on the Cu(II) removal and the corresponding flocculation capacities are shown in Figure 1d. The results signified that Cu(II) removal increased with the increasing dose of flocculant, reached a maximum at the optimal concentration of about 200 mg·L^−1^, and then decreased with a further increase in dose. The curves were representative of a typical flocculation system that is controlled by charge neutralization mechanisms [21]. Specifically, when the surface charges of the metal cations were completely neutralized by the addition of an anionic flocculant, the maximum Cu(II) removal efficiency was achieved. However, an increase in the Cu(II) residual content at flocculant overdosage could be observed, indicating that the colloid exhibits a re-stabilization phenomenon at higher flocculant doses in the presence of excess anionic charges. Three different Cu-doped carbon materials (SFC-x) were synthesized using flocculation sludge with different Cu(II) flocculation capacity (x = 0.9, 0.6, and 0.25 mg·mg^−1^).

### 3.2. Characterization of SFC-x

The structure of SFC-x was assessed by XRD (Figure 2a). The diffraction peaks of SFC-x emerged at 43.5°, 50.4°, and 74.2°, respectively, corresponding to the (111), (200), and (220) planes of Cu (JCPDS file No. 03-1015). Moreover, other characteristic peaks of SFC-0.9 located at 36.4° assigned to the (111) planes of Cu_2_O and more peaks of SFC-0.6 were evident, including peaks at 29.5°, 36.4°, 42.3°, and 61.3° assigned to the (110), (111), (200), and (220) planes of Cu_2_O, respectively (JCPDS No. 78-2076). These results suggested that Cu-doped carbon materials have been prepared successfully. XPS was employed to investigate the elemental states of SFC-x. Figure 2b shows the XPS survey spectrum of SFC-x; the photoelectron peaks for C 1s (284 eV), N 1s (400 eV), O 1s (531 eV), and Cu 2p (932.6 eV) were observed, respectively. From the XPS data, the Cu content in SFC-0.9 accounted for 21.61 at % of the specimen, which was much higher than that of SFC-0.6 (12.3 at %) and SFC-0.25 (8.5 at %) (Appendix A). These results demonstrated that the Cu content in SFC-x displayed a linear positive correlation with the flocculation capacity of Cu(II). According to a previous study, the electrochemical characteristics of metal-doped electrode materials are closely related to the metal content and metal species [23]. Relatively high metal content means more available active sites were exposed, leading to more redox reactions between the electro-active materials and the electrolyte, and high performance as an anode material for supercapacitors [24]. The deconvolution of the Cu(2p) peaks is shown in Figure 2c. SFC-0.9 shows major peaks of Cu 2p1/2 (952.5 eV) and Cu 2p3/2 (932.7 eV) besides the small peaks (944.20 eV), characteristic of Cu, suggesting that Cu-doped carbon materials have been prepared successfully. The C 1s XPS spectra of SFC-0.9 (Figure 2d) has three peaks of C=C–C (284.7 eV), C–N/C–O (285.6 eV), and C=O (288.1 eV) [25]. The C=C bonds of SFC-x accounted for 35.15 at % (x = 0.9), 36.69 at % (x = 0.6), and 37.07 at % (x = 0.25) of the content (Appendix A), respectively, which plays a key role in enhancing the electrochemical performance by improving electron mobility and lowering electrode resistance [26]. It is accepted that the XPS results were in agreement with those of XRD.

The BET surface area and pore size of SFC-x were estimated by N_2_ adsorption-desorption isotherms and Barrett-Joyner-Halenda (BJH) pore-size distribution analysis. The N_2_ adsorption isotherms of SFC-0.9 and SFC-0.6 displayed typical features of type IV isotherms with well-defined plateaus between P/P_0_ of 0.1~0.9 and an obvious hysteresis loop at the P/P_0_ >0.4 (IUPAC classification), which corresponds to the presence of mesopores (Figure 2e) [27]. Besides, the N_2_ adsorption isotherms of SFC-0.25 were type IV isotherms with an unapparent H1-type hysteresis loop [28]. Moreover, the specific surface area (S_BET_) and pore structure of SFC-x were surprisingly regulated by the flocculation capacity of Cu(II) in flocs (S_BET_ and pore structure parameters of SFC-x are presented in Appendix A). The S_BET_ of SFC-x was enlarged with the decreased copper flocculation capacity. The S_BET_ were 68.54, 258.48, and 285.24 m^2^·g^−1^ for SFC-0.9, SFC-0.6, and SFC-0.25, respectively. Notably, the fact that the SBET of SFC-0.9 was the smallest may be ascribed to the carbon content of SFC-0.9, which was the lowest (Figure 2b). The pore size distribution of SFC-x was assessed by means of the BJH model on the adsorption isotherm branches (Figure 2f). The pore size distributions of all samples were mainly in the range of 2 to 6 nm, which corresponded to the mesopore and was beneficial to the specific capacity via decreasing the ion transfer impedance from the electrolyte to micropores and inducing the electrical double layer formation [29]. Based on the suitable pore size distribution, SFC-x is expected to have superior capacitive performance.

Morphological features of SFC-x were analyzed through SEM and HRTEM. It was clear that the sizes and morphologies of SFC-x were dependent upon the Cu(II) flocculation capacity (heavy metal content) in flocs. As shown in Figure 3a, the SFC-0.6 has a cubic-like structure synthesized with Cu(II) flocculation capacity of 0.6 mg·mg^−1^ in flocs. A previous study has shown that porous Cu_2_O–Cu cubes can be prepared by reducing Cu(II) chelate at higher temperatures [30]. We also found that, as the Cu(II) flocculation capacity increased from 0.6 to 0.9 mg·mg^−1^ in sludge, the morphologies of SFC-0.9 changed from cubic to accumulated nano-cubes particles (Appendix A). In contrast, for SFC-0.25, we observed a change in morphology from cubic to a hierarchical flower-like structure with increasing Cu(II) flocculation capacity from 0.25 to 0.6 mg·mg^−1^ (Appendix A). The results indicated that the morphology of Cu-doped carbon materials can be easily tuned by the Cu(II) flocculation capacity in sludge.

The microstructure of the Cu-doped carbon materials was further investigated by HRTEM technique. From the insert in Figure 3b, lattice spacings of SFC-0.6 emerged at 0.208 nm and 0.246 nm, corresponding to the (111) crystal plane of Cu and the (111) plane of Cu_2_O. In addition, similar lattice fringes have also been observed from HRTEM image of SFC-0.9 and SFC-0.25 (Appendix A), with average size of lattice spacings of about 0.208 nm, which corresponds to the (111) lattice planes of the Cu structure. These results were consistent with the peaks of XRD, further indicating that Cu-doped carbon materials have been prepared successfully. Additionally, the elemental mapping images of SFC-0.6 revealed that the C, N, O, and Cu elements were uniformly dispersed (Figure 3c). As mentioned above, the main flocculation mechanism of Cu(II) was proposed on the basis of charge neutralization mechanisms, thus, highly dispersed Cu nanoparticles on the carbon framework were attained by combining the coagulation of Cu(II) and thermal treatment of the flocs. Uniform distribution of the Cu nanoparticles were easily accessed by the electrolyte ions as an electrode material, resulting in more activity towards redox reactions for supercapacitors [31].

### 3.3. Electrochemical Performance of SFC-x

The electrochemical performance of SFC-x was assessed via the three-electrode configuration in 6 M KOH. Figure 4a displays the CV curves of SFC-x at a potential scan rate of 5 mV·s^−1^ (potential window: −1 to 0 V). It was apparent that the area surrounded by the CV curves of the SFC-0.9 was larger than those of the SFC-0.6 and SFC-0.25 at the same scan rate, signifying that the SFC-0.9 possessed the highest specific capacitance. Besides, the curve shape had similarities and discrepancies from the rectangular shape controlled by the electrical double layer capacitance. The clear current peaks at −0.39 V and −0.12 V were ascribed to the oxidation of Cu^0^ to Cu^2+^ in the electrode [32]. In addition, two peak currents were evident when scanning from 0 V to −1 V (at −0.38 V and −0.82 V), ascribed to the reduction of Cu^2+^ to Cu^0^ in the electrode [33]. These results demonstrated that the pseudo-capacitance behavior of SFC-x was due to the transformations between Cu^0^ and Cu^2+^. From the redox peaks of copper in aqueous KOH electrolyte [34,35], the pseudo-capacitance behavior in the CV curves was associated with the following reactions (Equations (4) and (5)):(4)$$Cu_{2}O+2OH^{-}↔2\mathrm{CuO}+H_{2}O+2e^{-}$$
(5)$$2Cu+2OH^{-}↔Cu_{2}O+H_{2}O+2e^{-}$$

The CV measurements of SFC-0.6 under different scan rates are presented in Figure 4b. It was observed that the shape of the CV curves of SFC-0.6 was not obviously distorted with an increase in scan rates, indicating an ideal capacitive behavior. Obviously, the intensity of current peaks decreased upon increasing the scan rate from 5 to 50 mV s^−1^. Since the ions do not have sufficient time to diffuse into the minute pores and reach the active sites at higher scanning rates, which makes a very weak redox cycling in the electrode surface, thus leading to a decline in the current peaks. Furthermore, the GCD curves are a powerful measure for evaluation of supercapacitive performance. From GCD measurements of SFC-x at a current density of 1 A g^−1^ (Figure 4c), the specific capacitances of SFC-0.9, SFC-0.6, and SFC-0.25 were 637.9, 389.9, and 308.6 F·g^−1^, respectively. These results indicated that the flocculation capacity of Cu(II) in flocs was linearly positively related to the specific capacitance values of SFC-x. Moreover, it was evident that changes occurred twice in the slope of the potential in the discharge and charge curves, e.g., the slope changes occurred during S1, S2, S3, and S4 for SFC-0.9, which correlated with the current peaks observed in the CV curves. Compared with the ideal smooth charge/discharge curve, the charge/discharge curves of SFC-x display deviation further indicated that the SFC-x exhibits pseudo-capacitive behaviors. The GCD measurements of SFC-0.6 at various current densities are shown in Figure 4d. The specific capacitance values of SFC-0.6 were 427.1, 389.9, 346.9, and 187.4 F·g^−1^ that corresponded to the current densities of 0.5, 1, 2, and 5 A·g^−1^, respectively. Notably, the shape of the GCD curve was maintained up to 5 A·g^−1^, indicating a low equivalent series resistance [36]. A high capacitance of 187.4 F·g^−1^ could still be retained at a high current density of 5 A·g^−1^, which was greater than that of other Cu-based nanocomposites [37,38].

As shown in the Nyquist plot in Figure 5a, the spectra of the prepared samples were composed of a high-middle frequency region (semi-circle) associated with the charge-transfer at the electrode/electrolyte interface, and a low-frequency region (straight line) due to Warburg impedance [39]. The equivalent series resistance (ESR) of the SFC-x was acquired from the X-intercept of the sloping line at the low frequency region. Compared to SFC-0.6 and SFC-0.25, there was a maximum tilt angle line for SFC-0.9, indicating the lowest ESR for the latter. These results revealed that the conductivity and charge transfer efficiency of SFC–x could be significantly enhanced via augmenting the flocculation capacity of Cu(II). The charge transfer resistance (R_CT_) values of the electrodes can be represented by the radius of semicircle arcs on the x-axis, and a smaller semicircle represents a smaller charge transfer resistance [40]. It was also found that the R of SFC-0.9 was clearly lower than that of the others, indicating a fast charge transfer. The low resistance was chiefly attributed to the high conductivity and low resistance metal–metal contacts.

The specific capacitance of SFC-x as a function of current density is shown in Figure 5b (Calculated from GCD of SFC-x in curves of Figure 4d and Appendix A). The specific capacitance values of SFC-x decreased rapidly with an increase in current density, which could be attributed to the decrease in IR and the difficulty of the electrolyte ions in accessing the reactive sites of SFC-x for redox reactions at high current densities. The capacitance values of SFC-x were similar to, or greater than that of previously published reports on Cu nanoparticles [32], Cu_2_O/CuO/RGO [41], 3D porous CuO [42], and Cu_2_O/RGO [43]. Since the electrode materials were prepared from toxic sewage sludge, a byproduct of wastewater treatments, this process is crucial for economic and social development because it is beneficial with respect to both environmental and financial aspects.

Although the specific capacitance of SFC-0.9 was the highest among the samples, it has the lowest Cu(II) removal rate (R = 45.02%) which might have a detrimental impact on the environment. Figure 5c presents the Cu(II) removal efficiency and the possible energy storage benefits. With an increasing flocculant dosage, Cu(II) removal increased and the environmental impact was decreased, but there was a rapid decrease in Cu(II) removal with a further increase in flocculant dosage due to the re-stabilization of flocs. However, as a result of the low content of Cu supported on carbon materials, electrochemical performance of SFC-x gradually decreased with the increasing flocculant dosage. From a practical point of view, equilibrium should exist between environmental impact (heavy metals removal efficiency from wastewater) and energy (specific capacitance as a supercapacitor electrode).

These results suggested that the flocculant dosage up to 200 mg·L^−1^ is very promising in terms of comprehensive efficiency, at which more than 99% Cu(II) removal efficiency could be achieved, while the resulting annealed products (SFC-0.6) exhibited a high specific capacity (389.9 F·g^−1^ at 1 A·g^−1^). Also, long cycle life is an essential requirement of an electrode material for practical application in supercapacitors. The cycling performance of SFC-0.6 was assessed at high current densities of 5 A·g^−1^ for a 2500 charge-discharge cycle (Figure 5d). As expected, SFC-0.6 displayed high stability with only a 4% decline in specific capacitance after 2500 cycles. The high cycling stability could be ascribed to the cube-like structure and the large surface area of the porous structure, which mitigated the volume expansion during repeated charge/discharge phases. Two-electrode test cells are more closely related to the physical configuration and charge transfer for practical application of a supercapacitor, though lower values of capacitance are typically observed [44]. The electrochemical performance of SFC-0.6 was estimated from the GCD curves in a two-electrode system (Appendix A). An energy density as high as 7.6 Wh·kg^−1^ (at 0.5 A·g^−1^) was obtained, with a power density greater than 3400 W·kg^−1^ at a current load of 10 A·g^−1^. The superior supercapacitor performances of SFC-x were attributed to high Cu content supported on carbon, as well as the fully developed pore structures. Furthermore, the formation of Cu-doped carbon materials contributed to redox pseudo-capacitance during rapid charge and discharge cycles, which further enhanced the electrochemical performance of the electrode for supercapacitors. More importantly, the heavy metal would be confined within the carbon matrix by the complete pyrolysis of sludge, which effectively prevents the secondary heavy metal pollution and emissions from the leachate (Scheme 1).

## 4. Conclusions

In this work, a carboxyl group-functionalized starch derivative was synthesized and used as an effective flocculant for Cu(II) removal from wastewaters. Employing these Cu-contaminated sewage sludge as precursors, Cu-doped carbon materials were prepared as efficient electrode materials for supercapacitors through one-step carbonization without any additional metal salt. The results revealed that the specific capacitance of the resulting annealed products was linearly positively correlated to the Cu(II) flocculation capacity. With respect to the environmental capacity and energy capacity, the Cu(II) removal efficiency has been analyzed and compared to the possible energy benefits. The flocculant dosage up to 200 mg·L^−1^ was an equilibrium point existing between environmental impact and energy, as high as 99.50% Cu(II) removal efficiency could be achieved. Moreover, the resulting annealed product (SFC-0.6) exhibited a high specific capacity (389.9 F·g^−1^ at 1 A·g^−1^) and long cycling stability, with only 4% loss after 2500 cycles. This work presents a new approach to recycling heavy metal-contaminated sewage sludge to synthesize advanced energy storage materials, which is highly promising for commercial applications ranging from the energy to environment fields.

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
