# Peer review of "Cu-Doped Porous Carbon Derived from Heavy Metal-Contaminated Sewage Sludge for High-Performance Supercapacitor Electrode Materials"

_nanomaterials, 2019, doi:10.3390/nano9060892_

Reviewer 1 Report

The manuscript deals with the solution for the improvement of Cu removal from wastewater, pyrolytic conversion of sewage sludge to Cu-doped porous carbon and it using in energy storage system. The Authors presented a large number of references to support the initial literature review and the results section. However, some points in the introduction and methodological sections require revision by the Authors before publication (see comments below).

1. The introduction is written clearly and the object of researches is placed in a broad context. I suggest that Authors should expand some of issues and better emphasize the novelty of their work.

Lines 31: Please add other important sources of heavy metals.

Lines 37: Please define "large quantities" - add information about quantities of sewage sludge which are generated in the world or in your country.

Lines 41: Please add other important methods of sewage sludge management and briefly describe what quantities these waste are managed by each methods (in the world or in your country – respectively to lines 37).

Lines 62:  I suggest supplementing this part by adding brief description the most important conclusions concerning improvement of the electrochemical performance of carbon materials through metal-doping (Mn, Ni, Co, Fe).

2. Materials and methods: Authors should add more information, which allow to repeat the experiments.

Lines 101-107: Please add information about polyvinylidene fluoride and 1-methyl-2-pyrolidone. It is used in electrochemical measurements, but wasn’t described in part “Materials and methods”.

Lines 113: Please improve symbol of the water.

Lines 108-126: Please add comprehensive description how did you explore the effects of pH for the flocculation of Cu. In the part “Removal of Cu (II) from aqueous solution” (lines 161-185) Authors described results of this experiment, but the part “Materials and methods” doesn’t contain description of this studies.

Lines 135: Please add the instrumental operating conditions and analytical conditions (wavelength).

3. Results: Authors broadly described results and compared with previous studies. I have only two suggestion.

Lines 186-187: Please check the flocculant dosage and concentration of Cu (II). If I understood correctly, each aqueous solution had different volume (41.8 – 43.6 mL), not 40 mL. Therefore, in my opinion it should be 57-238 mg·L-1 of flocculant dosage and 110-114 mg·L-1 of Cu (II). Please check, if this calculations have meaning for the results.

Lines 200-204: This part should be moved to the introduction, since it motivates the work by the Authors and emphasizes the novelty.

4. Conclusions: The conclusions of the paper are clearly stated. It will be interesting to add the some future directions.

Lines 379-384: Please improve this part, because it is very similar to the abstract.

5. Did you perform the same experiment with urban wastewater, in the aim to study the matrix effect? The title of this work suggests that Cu came from sewage, but the Authors describe the experiment using commercially available Cu and distilled water. Please indicate, what are the flocculation capacity changes and what problems appear when Cu is removes from "real" wastewater.

Author Response

1. The introduction is written clearly and the object of researches is placed in a broad context. I suggest that Authors should expand some of issues and better emphasize the novelty of their work.

Lines 31: Please add other important sources of heavy metals.

Response: We appreciate the reviewer’s very useful suggestions. Other important sources of heavy metals have been added in line 31 paragraph 1, section 1: “(“mining, smelting activities, exhaust gas discharge and sewage irrigation”). Thanks!

Lines 37: Please define "large quantities" - add information about quantities of sewage sludge which are generated in the world or in your country.

Response: According to the referee’s suggestion, the information about quantities of sewage sludge has been added in the revised manuscript (line 38, paragraph 1, section 1: “(“It is estimated that over 8x 106 tons of sewage sludge are generated each day from wastewater treatment plants in China”)”). Thanks!

Lines 41: Please add other important methods of sewage sludge management and briefly describe what quantities these waste are managed by each methods (in the world or in your country–respectively to lines 37).

Response: According to the referee’s suggestion, we have added other sewage sludge management processes (Environ. Res. 2017, 156, 39-46), such as landfill, incineration processes, soil application and dumping at sea (line 43, paragraph 1, section 1). The situation of sludge disposal in China was illustrated in Fig. 1 (Water Res., 2015, 78:60-73). It suggests that the situation of sludge disposal in China is poor. More than 80% of sludge was dumped improperly, which caused serious secondary pollution. Generally speaking, sanitary landfill is the most commonly used method, followed by land application, incineration and building materials. We also add more discussion in line 45-47, paragraph 2, section 1 to better explain the different quantities proportion of each methods in China.

Fig. 1 The situation of sludge disposal in China (Water Res., 2015, 78:60-73).

Lines 62: I suggest supplementing this part by adding brief description the most important conclusions concerning improvement of the electrochemical performance of carbon materials through metal-doping (Mn, Ni, Co, Fe).

Response: We appreciate the reviewer’s question. We have added more discussion in line 66-69, paragraph 2, section 1 to better explain the role of metal-doping: “which can store and release energy reversibly through surface redox reactions and display considerably higher specific capacitance through an additional pseudocapacitive storage contribution between the electro-active materials and the electrolyte”.

2. Materials and methods: Authors should add more information, which allow to repeat the experiments.

Lines 101-107: Please add information about polyvinylidene fluoride and 1-methyl-2-pyrolidone. It is used in electrochemical measurements, but wasn’t described in part “Materials and methods”.

Response: More information about polyvinylidene fluoride and 1-methyl-2-pyrolidone has been added in line 109, paragraph 3, section 2.1 as highlighted. Thanks!

Lines 113: Please improve symbol of the water.

Response: We apologize for the typo. It has been corrected. Thanks!

Lines 108-126: Please add comprehensive description how did you explore the effects of pH for the flocculation of Cu. In the part “Removal of Cu(II) from aqueous solution” (lines 161-185) Authors described results of this experiment, but the part “Materials and methods” doesn’t contain description of this studies.

Response: The details were not described clearly in the previous manuscript, which was ambiguous to the readers. So, more detail had been added into experimental section of revised manuscript (line 129-132, paragraph 4, section 2.2): “Furthermore, 2.4 ml flocculant solution was added to a glass beaker (100 mL) containing 1.2 mL Cu(II) stock solution at pH 3, 3.5, 4, 4.5, 5, 5.5, 7 and 8.5, respectively. Water was added to obtain a total volume of 40 mL. After that, the same steps above was used to conduct the remaining flocculation experiments and detection of Cu (II).”

Lines 135: Please add the instrumental operating conditions and analytical conditions (wavelength).

Response: It has been added into experimental section of revised manuscript (line 149, paragraph 4, section 2.4). Thanks.

3. Results: Authors broadly described results and compared with previous studies. I have only two suggestion.

Lines 186-187: Please check the flocculant dosage and concentration of Cu (II). If I understood correctly, each aqueous solution had different volume (41.8–43.6 mL), not 40 mL. Therefore, in my opinion it should be 57-238 mg·L-1 of flocculant dosage and 110-114 mg·L-1 of Cu (II). Please check, if this calculations have meaning for the results.

Response: The details were not described clearly in the previous manuscript, which was ambiguous to the readers. In order to keep the initial metal concentration unchanged, 1.2 mL Cu(II) stock solution (4 g·L-1) was transferred into a glass beaker (100 mL), and then a prescribed volume comprised of 0.6 - 2.6 mL of flocculant solution was added. After that, water was added to obtain a total volume of 40 mL. The final metal concentration was then 120 mg L-1. Therefore, we modified the description of the flocculation process at line 121 on page 4 in revised manuscript: “Specifically, 0.6, 1.4, 1.6, 2, 2.4 and 2.6 mL flocculant solution was added respectively to a glass beaker (100 mL) containing 1.2 mL Cu(II) stock solution at pH 8.1. After that, water was added to obtain a total volume of 40 mL. Finally, Cu (II) concentration was 120 mg L-1.”

Lines 200-204: This part should be moved to the introduction, since it motivates the work by the Authors and emphasizes the novelty.

Response: According to the referee’s suggestion, this part has been moved to the introduction (Lines 89-93 paragraph 1, section 1).

4. Conclusions: The conclusions of the paper are clearly stated. It will be interesting to add the some future directions.

Lines 379-384: Please improve this part, because it is very similar to the abstract.

Response: According to the referee’s suggestion, we have modified our conclusions: “The flocculant dosage up to 200 mg·L-1 was an equilibrium point existing between environmental impact and energy, as high as 99.50% Cu(II) removal efficiency could be achieved. Moreover, the resulting annealed product (SFC-0.6) exhibited a high specific capacity (389.9 F·g-1 at 1 A·g-1) and long cycling stability, with only 4% loss after 2500 cycles.” (line 390-393).

5. Did you perform the same experiment with urban wastewater, in the aim to study the matrix effect? The title of this work suggests that Cu came from sewage, but the Authors describe the experiment using commercially available Cu and distilled water. Please indicate, what are the flocculation capacity changes and what problems appear when Cu is removes from "real" wastewater.

Response: The present research was to investigate the efficiency of our prepared samples for heavy metal ion removal from simulated wastewater. Flocculation of "real" wastewater will be done in our future work. In a general way, the more cations (Na+, K+, Ca2+) in the real wastewater will bind to the flocculated carboxyl groups, resulting in weakening the chelation of heavy metal cations by flocculation (Sep. Purif. Technol., 2016, 158(6):124-136). Therefore, it will take more flocculants to effectively remove Cu(II) in "real" wastewater.

Reviewer 2 Report

I have minor comments on this very good manuscript. In line 118, page 4, instead of thrice, an outdated word, more apropriate word should be used. 

Regarding the Figure 1b, the relation between zero potent and pH. In text autors say that surface charge changed from zero to -60 mV at pH 8.1, and from the image it is visible that it is less than -40mV for mentioned pH value.

Author Response

1. I have minor comments on this very good manuscript. In line 118, page 4, instead of thrice, an outdated word, more apropriate word should be used.

Response: According to the referee’s suggestion, "three times" instead of "thrice" and we apologize for the typos. It has been corrected in the revised manuscript (line 128, paragraph 4, section 2.2). Thanks.

2. Regarding the Figure 1b, the relation between zero potent and pH. In text autors say that surface charge changed from zero to -60 mV at pH 8.1, and from the image it is visible that it is less than -40mV for mentioned pH value.

Response: We apologize for the typos. “-60” should be “-38” and it had been corrected within the text. Thanks!

Reviewer 3 Report

The manuscript concerns the complete solution for removal of Cu from sewage sludge. The topic is interesting and research are important due the sustainable management with of such type of waste. The research are extensive and presented results are impressing, however the part concerning risk assessment is too general, there is no methods given, no results, thus there are no grounds to draw conclusions. Other minor remarks are marked in the attached file.

Author Response

The manuscript concerns the complete solution for removal of Cu from sewage sludge. The topic is interesting and research are important due the sustainable management with of such type of waste. The research are extensive and presented results are impressing, however the part concerning risk assessment is too general, there is no methods given, no results, thus there are no grounds to draw conclusions. Other minor remarks are marked in the attached file.

Author responses: We highly appreciate the critical comments given by reviewer.

1. Please add some results concerning evaluated risk (Lines 23-24).

Response: We highly appreciate the valuable comments given by reviewer. It would seem that the concept of “environmental risk” was misused in this paper. The potential risks of heavy metal pollution have not yet been taken into account in the experimental design nor in the analyses in this work. So, we modified our expression and deleted the concept of “environmental risk” in Line 22 on Page 1 in revised manuscript: “The Cu(II) removal efficiency have been evaluated and compared to the possible energy benefits.” The goal of wastewater treatment is to remove as many pollutants as possible in water bodies. To this end, it is necessary to find the optimal flocculation conditions in order to maximize Cu(II) removal efficiency and minimum discharge values of effluent discharge. In this study, the effects of flocculation conditions such as pH and dosage of flocculant on the Cu(II) removal efficiency were investigated, and the Cu(II) flocculation capacity was measured. Therefore, the expression of “The Cu(II) removal efficiency” seems more appropriate than “risks”. Moreover, we have checked the whole manuscript and revised the “environmental risk” in revised manuscript.

2.Toxicity depends on dose of Cu as it is the microelement. I would suggest to remove the expression "toxic"(Lines 84).

Response: It has been corrected in the revised manuscript. Thanks!

3.The same sentence in the abstract. Please add some information how was risk assessed (Lines 94).

Response: Please see the response to comment 1 from reviewer 3. We modified our expression and revised the “environmental risk” in line 101 on Page 3 in revised manuscript: “Finally, the Cu(II) removal efficiency has also been evaluated relative to possible energy storage benefits.”.

4. Lack of decription of methods for risk assessment (Lines 99).

Response: Please see the response to comment 1 from reviewer 3. We modified our expression and revised the “environmental risk” in Line 22 on Page 1 in revised manuscript “The Cu(II) removal efficiency has been evaluated and compared to the possible energy benefits.”

5. I would suggest to add legend that figure could state alone (Lines 217).

Response: According to the referee’s suggestion, the elemental analysis data of SFC-x were included in the supplementary information (Figure S3). It was clear that the Cu content in SFC-x displayed a linear positive correlation with the flocculation capacity of Cu (II).

Figure S3 The elemental analysis of SFC-x

6. How was results on environmental impact obtained? What is the dosage of flocculants on horizontal axis? What is the parameter on the vertical axis? (Lines 332-333).

Response: It is a good question raised by the reviewer. The result of the Cu(II) removal efficiency was summarized from Figure 1d. With an increasing flocculant dosage, Cu(II) removal was increasing and concentration of Cu (II) in the wastewater met national emission standards (Environ. Pollut., 2016:S0269749116305735), which was harmless to the environment. However, with a further increase in flocculant dosage, Cu (II) removal decreased due to the re-stabilization of flocs. High concentration of Cu (II) in the wastewater had detrimental effects on human health and environmental sustainability (J. Environ. Manage., 2011, 92(3):407-418), increasing the environmental impact. The dosage of flocculants on the horizontal axis was from 60 mg·L-1 to 260 mg·L-1. Figure 5c presents the Cu(II) removal efficiency and the possible energy storage benefits. Upon increasing the flocculant dosage, the parameter on the vertical axis was representing the changing trend of comprehensive efficiency, the Cu(II) removal efficiency and electrochemical performance. As for this type of the changing trend was represented by the vertical axis, it had appeared in many documents (Colorectal Dis., 2015, 13(3):312-316, Int. Bus. Rev., 2002, 11(6):685-705, Energy, 2014, 64:1026-1034). As above analysis, the red line representing the Cu(II) removal efficiency started to rise and then decline. Besides, the blue line representing electrochemical performance of SFC-x continuously declined, because the specific capacitance values of SFC-x were decreasing with an increasing flocculant dosage (Figure 4c). Combining the Cu(II) removal efficiency and electrochemical performance, the black line comprehensive efficiency showed increase at first and then decline. There is an equilibrium point between environmental impact and energy when the dosage of flocculant reach to 200 mg/L, at which more than 99% Cu(II) removal efficiency was achieved, while the resulting annealed product showed a high specific capacity (389.9·F·g-1 at 1·A·g-1) and good cycling stability (4% loss after 2500 cycles) as an electrode material for supercapacitors.

7. What is the methodology for stating this? (Lines 347).

Response: Please see the response to comment 1 from reviewer 3. We modified our expression and revised the sentence “An assessment of potential environmental risks” in line 341 on Page 12 in revised manuscript: “The Cu(II) removal efficiency and the possible energy storage benefits.”

8. This part is not decribed in the manuscript sufficiently. Lack of data for conclusions and this statement is to general to define it as environmental impact assessment or risk assessment (Lines 380).

Response: Please see the response to comment 1 from reviewer 3. We modified our expression and revised the sentence “the potential risks of heavy metal pollution” in line 389 on Page 13 in revise manuscript: “the Cu(II) removal efficiency has been evaluated and compared to the possible energy benefits.” Moreover, we have checked the whole manuscript and revised the “environmental impact assessment or risk assessment”.